# Toll-Like Receptor 2 at the Crossroad between Cancer Cells, the Immune System, and the Microbiota

**DOI:** 10.3390/ijms21249418

**Published:** 2020-12-10

**Authors:** Antonino Di Lorenzo, Elisabetta Bolli, Lidia Tarone, Federica Cavallo, Laura Conti

**Affiliations:** Department of Molecular Biotechnology and Health Sciences, University of Torino, 10126 Torino, Italy; antonino.dilorenzo@unito.it (A.D.L.); elisabetta.bolli@unito.it (E.B.); lidia.tarone@unito.it (L.T.); federica.cavallo@unito.it (F.C.)

**Keywords:** Toll-like receptor 2, cancer, combined therapies, tumor microenvironment, DAMPs, tumor microbiota

## Abstract

Toll-like receptor 2 (TLR2) expressed on myeloid cells mediates the recognition of harmful molecules belonging to invading pathogens or host damaged tissues, leading to inflammation. For this ability to activate immune responses, TLR2 has been considered a player in anti-cancer immunity. Therefore, TLR2 agonists have been used as adjuvants for anti-cancer immunotherapies. However, TLR2 is also expressed on neoplastic cells from different malignancies and promotes their proliferation through activation of the myeloid differentiation primary response protein 88 (MyD88)/nuclear factor kappa-light-chain-enhancer of activated B cell (NF-κB) pathway. Furthermore, its activation on regulatory immune cells may contribute to the generation of an immunosuppressive microenvironment and of the pre-metastatic niche, promoting cancer progression. Thus, TLR2 represents a double-edge sword, whose role in cancer needs to be carefully understood for the setup of effective therapies. In this review, we discuss the divergent effects induced by TLR2 activation in different immune cell populations, cancer cells, and cancer stem cells. Moreover, we analyze the stimuli that lead to its activation in the tumor microenvironment, addressing the role of danger, pathogen, and microbiota-associated molecular patterns and their modulation during cancer treatments. This information will contribute to the scientific debate on the use of TLR2 agonists or antagonists in cancer treatment and pave the way for new therapeutic avenues.

## 1. Introduction

The immune system provides a complex cellular and molecular protection from pathogen infection. In mammals, it is divided in two main branches, the innate and the adaptive immunity. The innate immune system is the first line of defense, while the adaptive immunity acts in later phases to eliminate the pathogen and generate an immunological memory. Thanks to the clonal selection of lymphocytes endowed with antigen-specific receptors, generated via mechanisms of gene rearrangement, the acquired immune system is able to trigger a highly specific immune response. Instead, the innate immune response is almost completely not specific, although it is able to distinguish self from non-self [1]. Despite the lack of specificity, the innate immune system plays an essential role in host defense, since it can recognize microbial-derived antigens through the pattern recognition receptors (PRRs) expressed on professional antigen-presenting cells (APCs), such as macrophages or dendritic cells (DCs). These cells are able to phagocyte the pathogens, process the foreign antigens, and load the derived peptides on major histocompatibility complex (MHC) molecules in order to present them to T lymphocytes and initiate a specific adaptive immune response [2].

The role of the immune system is not limited to the fight to infectious diseases, and both innate and acquired immunity are involved in the surveillance of endogenous alterations that might lead to pathological conditions, such as in the case of cancer. Tumor formation and progression are the result of a complex interplay between cancer cells and immune cells, in which the role of the immune system is not always what it might be expected. Indeed, if on one hand the immune system can activate responses able to hinder cancer progression, on the other hand, it has been demonstrated that cancer cells are able to evade the immune surveillance and to manipulate immune cells to make them exert a pro-tumor activity. Indeed, cancer cells can release or express on their surface immunosuppressive signals, and they can stimulate the activation and proliferation of T regulatory cells (Tregs), which in turn have a modulatory action on cytotoxic T lymphocytes (CTLs). Furthermore, through the maintenance of an inflammatory tumor microenvironment (TME) and the release of the appropriate factors, cancer cells can educate innate immune cells to exert tumor-protective and immunosuppressive activities [3]. The dichotomous role played by the innate immune system in cancer is exemplified by the ambivalent activity that PRRs, and in particular Toll-like receptor 2 (TLR2), exert in tumor progression [4]. TLR2 is crucial for the detection of danger signals released by cancer cells and for the activation of the innate as well as of the specific anti-tumor immune response [5]. However, in some cases, TLR2 activation can lead to an immunosuppressive effect that favors tumor progression. Moreover, when expressed on cancer cells, TLR2 activation may lead to cancer cell survival, proliferation, and invasion [6]. In this paper, we will review the controversial role played by TLR2 in cancer progression and discuss the possibility of TLR2 targeting for the development of more effective anti-cancer combined treatments.

## 2. TLRs

The discovery of TLRs had a huge impact in the field of innate immunity that led to two Nobel Prizes in Physiology or Medicine in 1995 to Christiane Nüsslein-Volhard, who discovered the Toll gene in Drosophila, and in 2011 to Jules Hoffmann and Bruce Beutler, who demonstrated that Toll in Drosophila and TLRs in mammals are involved in anti-fungal and innate immunity, respectively [7,8].

TLRs are one of four major families of PRRs, which include also NOD-like receptors (NLRs), RIG-like receptors (RLRs), C-type lectin receptors (CLRs), and represent the cornerstone of the innate immune response [9]. PRRs are transmembrane proteins associated to the cellular and endosomal membranes or the cytosol. They recognize molecules frequently associated with pathogens, the so-called pathogen-associated molecular patterns (PAMPs), leading to activation of downstream signal transduction pathways, which result in the production of inflammatory cytokines, type I interferons (IFNs), and other mediators. These processes trigger inflammation and regulate antigen-specific adaptive immune responses, and thus, they are essential for the clearance of infecting microbes [10]. In addition to PAMPs, PRRs are also involved in the recognition of damage-associated molecular patterns (DAMPs), as proposed for the first time by Polly Matzinger with the “Danger Theory”. Matzinger demonstrated that during tissue stress or damage, endogenous molecules are released or activated that induce the inflammatory response, which empowers antigen-presenting cells to activate the adaptive immune response [11]. Moreover, the innate immune system is capable of detecting perturbations induced by infections and other pathologic conditions in cell homeostasis through the recognition of homeostasis-altering molecular processes (HAMPs) [12].

Until now, 10 human and 13 murine subtypes of TLR have been identified, even though TLR10 is non-functional in the mouse. They are expressed on all innate immune cells including macrophages, Natural Killer (NK) cells, DCs, monocytes, and neutrophils as well as on adaptive immune cells such as T and B lymphocytes. Strikingly, the expression of TLRs is not restricted to immune tissues but is distributed in all tissues including heart, liver, colon, small intestine, pancreas, lung, kidney, skeletal muscle, brain, ovary, placenta, testis, and prostate, where TLRs can also be detected on non-immune cells such as epithelial and endothelial cells and fibroblasts [4,13]. TLRs are type I transmembrane proteins endowed with 20–27 extracellular leucine-rich repeats for the recognition of PAMPs/DAMPs, transmembrane domains, and intracellular toll-interleukin 1 (IL-1) receptor (TIR) domains responsible for the activation of downstream signal transduction pathways [14].

TLRs are functionally subdivided into cell membrane TLRs (TLR1, TLR2, TLR4, TLR5, TLR6, and TLR10) and intracellular TLRs or nucleic acids sensors (TLR3, TLR7, TLR8, and TLR9) that are localized in the endoplasmic reticulum (ER), endosomes, and lysosomes [15].

Cell surface TLRs mainly recognize microbial membrane components such as proteins, lipids, and lipoproteins, or endogenous DAMPs produced following injury and non-physiological cell death, including elements of damaged/fragmented organelles, extracellular matrix components, such as hyaluronan and fibrinogen, plasma membrane constituents, nuclear and cytosolic proteins, among which high-mobility group box protein 1 (HMGB1) and heat shock proteins (HSPs). Intracellular TLRs recognize nucleic acids derived from bacteria and viruses and also self-nucleic acids in disease conditions such as autoimmunity [16].

Once TLRs bind PAMPs/DAMPs, a signaling cascade is activated by the interaction between the TIR domain and other adaptor molecules, such as myeloid differentiation primary response protein 88 (MyD88), TIR domain containing adaptor inducing IFN-β (TRIF), TIR domain-containing adaptor protein (TIRAP or MAL), TRIF-related adaptor molecule (TRAM), Sterile α- and armadillo-motif-containing protein (SARM). There are two different TLR signaling pathways depending on the adaptor molecule recruited: the MyD88-dependent pathway and the MyD88-independent, also called TRIF-dependent, pathway. These pathways converge on the activation of mitogen-activated protein kinase (MAPK), activating protein-1 (AP-1), nuclear factor kappa-light-chain-enhancer of activated B cells (NF-κB), or interferon regulatory factors (IRF), which finally lead to the production of pro-inflammatory cytokines, such as tumor necrosis factor alpha (TNF-α), IL-1β and IL-6, type I IFNs, chemokines such as CXCL8 and CXCL10, and antimicrobial peptides. All TLRs, except TLR3, utilize the MyD88-dependent pathway to induce the production of inflammatory cytokines, while the TRIF-dependent pathway used by TLR3 and TLR4 results in the stimulation of type I IFNs [16,17] (Figure 1).

## 3. TLR2

TLR2, together with TLR1, TLR3, TLR4, and TLR5, was first identified and characterized in 1998 [18]. TLR2 is the only TLR that forms functional heterodimers with more than two other types of TLRs, forming dimers with TLR1, TLR6, and in some cases with TLR4. TLR2 interacts also with different non-TLR molecules, allowing the recognition of a great variety of PAMPs from all microbial phyla including viruses, fungi, bacteria, and parasites. In particular, TLR2 can sense highly conserved lipoproteins expressed on the outer membranes of Gram-positive bacteria and, in association with its co-receptor CD14, membrane antigens of some Gram-negative bacteria, such as lipopolysaccharide (LPS). Independently from the dimers formed, the downstream signaling cascade proceeds through the MyD88-dependent pathway, leading to cytokine production [19].

TLR2 is expressed by immune cells, endothelial cells, and epithelial cells. This wide expression is consistent with the widespread range of roles and functions of TLR2 [20]. A beneficial role for TLR2 signaling is described in the maintenance of mucosal homeostasis and defense against some pathogens, whereas TLR2 signaling stimulated by other pathogens or after endogenous activation is correlated with a more severe phenotype in infectious and inflammatory diseases. In addition, there is a clear role for TLRs in microbe–host and host–microbe interactions, thus mediating the cross-talk with tissue microbiota [21]. Moreover, TLR2 activation by DAMPs released from damaged tissues mediates a kind of inflammatory response. For example, TLR2 can be activated by extracellular matrix components, cytoplasmic proteins as S100 or HSPs, and nuclear structural proteins such as histones and HMGB1 [9]. Thus, TLR2’s ability to interact with several self or non-self molecules confers it a key role not only in anti-pathogens response but also in the maintenance of the homeostatic conditions. Several types of cells in the body can dialogue through TLR2 sensing and activation. The correct interplay between tissue cells, immune cells, and microbiota is a necessary condition for a healthy organism. However, this equilibrium can be lost, for example as a consequence of aggressive therapies (i.e., antibiotics or chemotherapy), microbiota alterations, or other disorders, such as cancer [21]. In that case, TLR2 may have a dual role and be a central element in the anti-tumor immune response or in the pro-tumorigenic processes. A deeper comprehension of these two mechanisms is essential for the development of new approaches for the design of anti-cancer therapies.

## 4. The Role of TLR2 in Anti-Tumor Immune Response

Lymphoid and myeloid cells present in the TME can have either an immune suppressive or immune stimulatory function, being therefore important regulators of cancer progression, survival, as well as therapy resistance [22]. The presence of dying cells in the TME causes the release of DAMPs, which in turn activate myeloid cells thanks to their PRRs [23]. For instance, HMGB1, a nuclear protein that upon cell damage or death can be released in the TME, is able to stimulate several PRRs signaling, including TLR2 [24]. DCs can be activated by DAMPs together with other stimuli, such as TNF-α and type I IFNs. Once stimulated, DCs enhance their antigen-presenting activity by increasing the expression of MHC class II and costimulatory molecules, resulting in the activation of an adaptive immune response [25]. To study whether TLR2 activation may promote anti-cancer immune responses, exogenous TLR2 ligands were administered in several murine cancer models. In lung cancer models, the administration of synthetic or bacterial-derived TLR2 ligands induces the differentiation of monocytic myeloid derived suppressor cells (M-MDSC) toward the anti-tumoral M1 phenotype, with the production of IFN-γ, nitric oxide (NO), and pro-inflammatory cytokines such as TNF-α, IL-12p40, and IL-12p70, suggesting a correlation between TLR2 activation and a better anti-tumor response [26,27]. The efficacy of TLR2 stimulation on tumor-infiltrated leukocytes was demonstrated also in a melanoma model. Indeed, it was reported that the administration of the synthetic compound diprovocim activates the TLR1/2 signaling and synergizes with ovalbumin (OVA) immunization and anti-programmed death-ligand (PD-L)1 treatment in mice engrafted with B16-OVA, promoting leukocyte infiltration in tumor as well as an increased systemic humoral and cytotoxic response [28]. Contrasting results were observed in a study on fibrosarcoma in vivo. Mice immunized with tumor-associated antigens (TAAs) together with the synthetic lipoprotein FSL-1, a TLR2/6 ligand, showed a significant decrease of tumor growth due to the induction of T helper type 2 (Th2) cells. This combined treatment produced an increase in tumor-specific CTLs and a humoral response able to mediate antibody-dependent cell-mediated cytotoxicity (ADCC), tumor lysis by complement activation, and a reduction of Treg cell frequency. However, the treatment with FSL-1 alone increased the number of Tregs and enhanced tumor growth. This effect was reverted by the combined administration of anti-CD25 antibodies, suggesting the involvement of Tregs in the FSL-1 pro-tumoral effect. None of these effects was observed in TLR2^−/−^ mice, confirming that both pro- and anti-tumor effects are TLR2-dependent [29]. TLR2 seems to play a role also in NK cell-mediated anti-tumor immunity. Indeed, in lung cancer-bearing mice, TLR2 stimulation by treatment with the natural polysaccharide SEP induces NK cell activation, proliferation, cancer cell-directed cytotoxicity, and the release of IL-2 and IFN-γ [30]. Furthermore, TLR2 stimulation with the natural polysaccharide krestin (PSK) activates human NK cells, inducing IFN-γ secretion and the lysis of K562 cells. In addition, PSK also enhances the efficacy of trastuzumab on SKBR3 cells as well as on Her2/Neu transgenic mice by potentiating the ADCC by NK cells [31].

However, although the role of TLR2 in tumors has been widely studied, it remains ambiguous. Indeed, there is a huge variation on the results obtained by different groups, depending on several factors, such as the type of tumor and the model used, with reports suggesting that TLR2 signaling can activate an anti-tumor immune response, while others highlight its immune suppressive function, as described in the next sections.

## 5. The Role of TLR2 in Immunosuppression

As discussed in the previous paragraph, TLR2 plays an important role in the activation of innate immune cells and may promote anti-tumor responses in virtue of its ability to activate antigen presenting cells and, consequently, tumor-specific T lymphocytes. However, TLR2 is also expressed on immune cells, such as Tregs, MDSCs, macrophages, and neutrophils, which can contribute to tumor growth and metastatic dissemination by stimulating the generation of an immunosuppressive microenvironment and of the metastatic niche [32]. Of note, TLR2 activation promotes their immunosuppressive function, thus favoring tumor growth [6] (Figure 2).

Since the characterization of the role exerted by TLR2 on immunosuppressive cell populations may provide therapeutic tools for the treatment of both autoimmune and neoplastic diseases, many researchers have investigated the effects of TLR2 signaling in MDSCs. These important studies demonstrated that TLR2 activation drives the differentiation of MDSCs from bone marrow progenitor cells and promotes their survival and proliferation by activating NF-κB and MAPK and inducing the production of IL-6 and the consequent activation of the signal transducer and activator of transcription (STAT3). Therefore, TLR2 activation increases MDSC accumulation in the tumor microenvironment and in lymphoid organs [33]. The same group of researchers subsequently demonstrated that TLR2 activation may promote the immunosuppressive activity of monocytic-MDSCs, inducing their differentiation into tumor-promoting macrophages. The underlying mechanism is very intriguing, since it involves the collaboration of tumor-infiltrating CD8^+^ T cells, which are the final victims of MDSC function. Indeed, MDSC-derived macrophages present antigenic peptides to CD8^+^ T cells, inducing their transient activation and the production of IFN-γ. Then, this cytokine leads to inducible nitric oxide synthase (iNOS) expression in macrophages, and iNOS-generated NO inhibits T cell proliferation induced by DCs [34]. Furthermore, TLR2 activation stimulates MDSCs to produce IL-10, which induces macrophage polarization toward the M2 tumor-promoting phenotype, while MDSC-produced IL-6 promotes tumor metastatic growth [35,36].

The immunosuppressive effect of TLR2 activation is not limited to MDSCs but takes place also in DCs. In fact, in the TME, cancer cells promote TLR2 activation in DCs, which induces the secretion of elevated levels of cytokines including IL-6 and IL-10 and the overexpression of their receptors. Thus, these cytokines act in an autocrine manner to activate STAT3 in DCs and reduce the expression of IL-12, MHC class II, CD40, and CD86. These dysfunctional DCs lose the ability to activate anti-cancer cytotoxic T cells while promoting the function of Treg cells [37]. These results suggested that TLR2 could indirectly influence the activity of Treg through a DC-mediated mechanism but did not exclude the presence of Treg cell-intrinsic effects. This gap was filled by a very interesting paper by Sutmuller et al. that, analyzing the direct effects induced by TLR2 agonists on Treg and T helper cells in vitro, showed that TLR2 activation in Treg cells induces their proliferation and may directly stimulate their immunosuppressive activity [38]. Physiologically, this mechanism is activated by commensal bacterial components of the microbiota that mediate the TLR2-dependent conversion of CD4^+^ T cells into forkhead box P3 (Foxp3)^+^ Treg cells that produce IL-10, with the aim of inducing mucosal tolerance [39]. These studies demonstrated the importance of TLR2 in immune system homeostasis and in the balance between tolerance and immunity, but no indication of the consequences that this immunosuppressive mechanism may have on tumorigenesis and cancer progression were provided. Therefore, several research groups addressed the question whether, in the tumor context, TLR2 can exert detrimental effects, promoting the suppression of effector T cell function. A seminal paper published in 2019 by the group of Prof. Wang demonstrated that secretory autophagosomes containing the TLR2 ligand HSP90α, which is released by cancer cells as a result of an alternative non-degradative autophagy mechanism used for the unconventional secretion of small molecules, stimulate IL-6 production in CD4^+^ T lymphocytes through the TLR2–MyD88–NF-κB axis. IL-6 promotes the production of IL-10 and IL-21 in an autocrine manner, conferring regulatory and immunosuppressive activity to T cells [40]. These data are very important, since they clearly demonstrate that cancer cells can induce TLR2 activation to interfere with host anti-tumor immunity. However, the role of TLR2 on Treg function is controversial, since other researchers demonstrated that TLR2 activation on Tregs transiently inhibits their immunosuppressive activity [38]. In our opinion, these data are not in contrast but reflect the multifaceted nature of TLR2, whose effects might be context-dependent.

In addition to myeloid and T cells, TLR2 is likewise expressed on B lymphocytes and induces their differentiation toward the T cell Ig and mucin domain (TIM)-1 expressing regulatory B cells (Bregs) [40] through the activation of the MAPK pathway. These Bregs produce high amounts of IL-10 and play a key role in the inhibition of effector T cell function and proliferation. A high infiltration of TIM-1^+^ B cells correlates with advanced disease stage and poor overall survival in patients with hepatocellular carcinoma [41].

The activation of TLR2 on tumor infiltrating myeloid or Treg cells is induced directly by cancer cells, which are able to secrete TLR2 endogenous ligands, such as the chondroitin sulfate proteoglycan versican, HMGB1 (which will be discussed in the following paragraphs) and Wnt5a [35,42]. This intercellular communication does not only take place in the TME, as tumor cells can influence immune cells at distant sites. Indeed, tumor cells secrete exosomes and secretory autophagosomes carrying endogenous TLR2 ligands (HMGB1, HSP72, HSP90, palmitoylated proteins and others) [40], which reach distant organs and, by activating the TLR2–IL-6–STAT3 signaling pathway in myeloid cells, promote the establishment of the pre-metastatic niche and contribute to the “seed and soil” mechanism responsible for metastasis development [43]. The pro-metastatic activity of TLR2 is confirmed by the fact that TLR2 knockout mice develop fewer lung, liver, and adrenal metastases following the subcutaneous implantation of cancer cells as compared to their wild-type counterpart [35,42].

Thus, we can conclude that TLR2 activation on immunosuppressive cells plays an important role for the maintenance of immune system homeostasis in physiological conditions but can exert detrimental effects when this happens in the context of carcinogenesis. Further studies are needed to verify whether the PAMP/DAMP/TLR2 axis may represent a safe and effective target for cancer treatment.

## 6. TLR2 Is Expressed on Cancer Cells and Exerts Pro-Tumoral Effects

Neoplastic cells from several hematological malignancies express TLR2, as their normal cell of origin did, and exploit its signaling pathway to promote their proliferation. In particular, TLR2 is overexpressed in the neoplastic CD34^+^ hematopoietic stem and progenitor cells characteristic of the myelodysplastic syndromes (MDS), and its activation induces the production of IL-8 and inhibits their erythroid differentiation [44]. Recent reports indicated that TLR2, as many other TLRs, is expressed on neoplastic cells from several solid tumors, such as colon, gastric, breast, pancreatic and oral cancers, where its activation promotes cancer progression and metastasis through different cell-intrinsic mechanisms [45,46,47]. The first indication that TLR2 is expressed on solid tumors came from a study published in 2002 by Yoshioka and colleagues, who demonstrated that different colon cancer cell lines express TLR2 and CD14 and, in response to LPS treatment, secret transforming growth factor (TGF)-β and hepatocyte growth factor (HGF) [48]. Subsequent studies demonstrated that TLR2 is expressed at low levels in normal colorectal epithelium, with a gradient of expression from the crypt bases to the upper parts of the crypts and the surface [49]. In colorectal cancer, TLR2 is expressed at high levels by most tumor cells [49], which is often as a consequence of the loss of the hypermethylated in cancer 1 (HIC1) tumor suppressor gene that normally inhibits its transcription [50]. It is interesting to note that the TLR2 expression pattern may differ between normal and neoplastic cells. Indeed, normal mucosal cells display a polarization of TLR2 expression in the basolateral membrane, whereas cancer cells display a diffuse TLR2 expression in the basolateral and apical membrane, as well in the cytoplasm [51]. This altered expression pattern is probably due to the downregulation of the ubiquitin-binding protein Toll-interacting protein (TOLLIP) in tumor lesions. In the normal mucosa, TOLLIP promotes the traffic of TLR2 into endosomes and hence to the basolateral membrane, and it may induce its early degradation, thus maintaining its low expression. On the contrary, in neoplastic cells, the decrease of TOLLIP results in high and diffuse TLR2 expression [51], which may increase the pro-tumorigenic signals activated by TLR2. In fact, TLR2 activation enhances reactive oxygen species (ROS) production in colon cancer cells and consequently activates sterol regulatory element-binding protein-2 (SREBP2), a transcription factor involved in cholesterol biosynthesis and uptake. The increased intracellular cholesterol levels eventually result in the promotion of cell proliferation [52]. As a consequence, TLR2 knockout or knockdown inhibits the proliferation of colorectal cancer cells, leading to cell cycle arrest in the G1 phase and thereby significantly impairing the development of both inflammation-related and sporadic colorectal cancer in preclinical models [53]. Of note, TLR2 is also implied in the development of chemoresistance in colorectal cancer. Indeed, TLR2 expression is further enhanced in colon cancer cells resistant to chemotherapy drugs such as 5-fluorouracil and oxaliplatin. Activation of the TLR2/6 heterodimer reduces the expression of miR-125b-5p, an miRNA that controls epithelial to mesenchymal transition (EMT) and the expression of drug resistance-related proteins, thus resulting in the acquisition of drug resistance and in the promotion of cell migratory and invasive activity [54]. Hence, high TLR2 expression is significantly associated with poor overall survival in colon cancer patients [46]; a trend of association of weak TLR2 expression in the invasive front and in lymph node metastases and better disease-free survival was observed [49]. The association between the TLR2 signaling pathway and colon cancer development is further supported by the presence of genomic amplifications of the TLR2 downstream effectors IL-1 receptor-associated kinase (IRAK)1 and IRAK4 in 18% and 39.7% of colon cancer patients, respectively. Moreover, nonsense mutations in the TLR2 gene that lead to the production of constitutively active truncated forms have been observed, although they are rare, and an increased risk of developing cancer is associated to some TLR2 single-nucleotide polymorphism (SNPs) [46,55].

Several recent studies support a role for TLR2 in promoting tumorigenesis, although the mechanisms involved in its regulation may differ among cancers. For instance, TLR2 is overexpressed in more than 50% of human gastric cancers due to the hyperactivation of the STAT3 oncogene that directly induces TLR2 transcription, and increased STAT3 activation and TLR2 expression correlate with poor prognosis in gastric cancer patients [56]. Indeed, high TLR2 expression is significantly associated with advanced disease stage, the presence of distant metastasis and microvascular invasion, and large tumor size [47]. TLR2 promotes gastric cancer progression by stimulating proliferation and inhibiting the apoptosis of cancer cells through the activation of the NF-κB, PI3K/AKT, ERK1/2, and JNK MAPK pathways, without inducing an inflammatory response [56]. A TLR2-dependent gene signature associated with increased cell growth and poor overall survival has been identified in human gastric cancer, corresponding to the up-regulation of six anti-apoptotic genes (BCL2A1, BCL2, BIRC3, CFLAR, IER3, TNFAIP3) and the down-regulation of the tumor suppressors PDCD4 and TP53INP1 [57]. These studies demonstrated the importance of TLR2 in tumor proliferation but lacked an investigation of its role in cancer cell metabolism. This topic was addressed by a very important paper demonstrating that TLR2 participates in the metabolic reprogramming of neoplastic gastric cells, enhancing both oxidative phosphorylation (OXPHOS) and glycolysis, with a bias toward glycolytic activity, thus fueling their high rate proliferation. This metabolic reprogramming increases lactate production but also compensatory protective antioxidant mechanisms, mediated by the TLR2-dependent up-regulation of the manganese-dependent superoxide dismutase (SOD)2, which is a mitochondrial antioxidant enzyme that controls oxygen and hydrogen peroxide production from superoxide anion radicals formed by OXPHOS, thus maintaining intracellular redox balance [47].

TLR2 plays a pivotal role in the progression of other solid tumors. For example, TLR2 induces a cancer cell-intrinsic stimulation of the proliferation of hepatocarcinoma cells in an infection-associated mouse model [58]. Moreover, it contributes to Hepatitis B Virus (HBV)-mediated liver tumorigenesis, since the Hepatitis B surface antigen (HBsAg) upregulates TLR2 on liver cells and induces their proliferation and invasion through TLR2/MyD88/NF-κB-mediated production of pro-inflammatory cytokines [59]. In prostate cancer, TLR2 is expressed on cancer cells and mediates the pro-tumorigenic effects induced by some pathogens such as *Trichomonas vaginalis* and *Mycoplasma* through the induction of IL-6 and IL-8 production and consequent stimulation of EMT [60,61]. Similarly, TLR2 is expressed on pancreatic cancer, and its activation induced by its endogenous ligands, such as pancreatic adenocarcinoma upregulated factor (PAUF), induces cell proliferation, migration and angiogenesis through the production of pro-tumorigenic cytokines, vascular endothelial growth factor (VEGF), and platelet-derived growth factor (PDGF) [62,63].

The relevance of the TLR2 signaling pathway in breast cancer is demonstrated by the fact that TLR2 expression has been found in human breast cancer samples, and that its expression is associated with poor overall survival and to resistance to endocrine therapy [64,65]. Moreover, multiple genetic alterations that lead to increased TLR2 signaling have been identified in human breast cancer specimens. Among these, are amplifications of the gene coding for IRAK1, which is found in 23.8% of breast cancers, and mutations producing constitutively active forms of TLR2 [46]. TLR2 expression is particularly high in breast cancer cell lines endowed with metastatic potential, and its activation induces invasiveness through the secretion of IL-6, TGF-β, VEGF, and the metalloproteinase (MMP)9, which degrades the extracellular matrix [66].

The majority of the papers analyzing the role of TLR2 in cancer progression were focused on the TLR2 cancer cell-intrinsic role and did not extensively investigate the role of TLR2 in the non-immune TME. However, two papers demonstrated that TLR2 also contributes to tumor angiogenesis in a VEGF-independent manner. Indeed, TLR2 is expressed on endothelial cells and promotes their proliferation and migration and a strong secretion of granulocyte-macrophage and granulocyte colony-stimulating factor (GM-CSF and G-CSF) [67,68].

Overall, the studies reported in this section demonstrate that TLR2 promotes tumor progression through cancer cell-intrinsic mechanisms, independently from its role in inflammation. However, albeit a role for TLR2 in the promotion of cancer angiogenesis was demonstrated, we think that a comprehensive analysis of the role played by TLR2 in the complex interplay between cancer cells and the heterogeneous cell populations present in the TME is still missing. This would represent a fundamental information for the development of TLR2-targeting anti-cancer therapies.

## 7. TLR2 Promotes Cancer Stem Cell Self-Renewal

Recently, we have demonstrated that TLR2 is expressed on cancer stem cells (CSCs), which are a small population of cells at the apex of tumor cell hierarchy. CSCs are characterized by self-renewal potential and by the ability to differentiate to give rise to the different cell types that compose the bulk of the tumors, and they have been implied in tumor onset, metastatic spreading, and resistance to current therapies [69,70,71,72]. We have previously demonstrated that breast CSCs express TLR2 and that its stimulation induces the activation of the MyD88/NF-κB and AKT pathways, which induces the production of IL-6, TGF-β, and VEGF. Then, these factors act in an autocrine/paracrine manner to activate STAT3 and Smad3 signaling pathways [73,74] (Figure 3). IL-6 induces EMT, thus increasing the CSC pool by promoting the transformation of more differentiated cancer cells into CSCs. Moreover, IL-6 recruits mesenchymal stem cells and immune cells in the TME, favoring the maintenance of an inflammatory milieu that promotes tumor growth [22]. Similarly, TGF-β induces EMT and the secretion of matrix components that stimulate invasion and metastatic spreading, and, together with VEGF, it recruits endothelial cells and promotes their proliferation, favoring angiogenesis [22]. Overall, TLR2 activation stimulates CSC survival, proliferation, and invasion [73]. Of note, breast CSCs secrete high levels of G-CSF as compared to more differentiated cancer cells (Figure 3, insert). G-CSF induces TLR2 expression [75,76], whose activation can further increase G-CSF production [68], thus generating an autocrine loop sustaining TLR2 expression in breast CSCs. Therefore, TLR2 silencing with specific siRNA significantly impairs tumor growth and prevents the development of lung metastasis in preclinical mouse models of Her2^+^ breast cancer [73]. Similarly, Scheeren et al. demonstrated that the TLR2/MyD88 signaling pathway plays a cell-intrinsic role in the function of mammary and intestinal stem cells, and it promotes the development of intestinal and ER negative breast cancers in preclinical models [46]. Of note, they demonstrated an additional mechanism of action of TLR2 activation, which was based on the TLR2/MyD88-mediated activation of the Wnt pathway, with the induction of the expression of its targets CD44 and Lgr5, which may lead to tumor initiation [46]. Recently, other groups reported the involvement of TLR2 in the promotion of CSC self-renewal in solid tumors. For example, Chen et al. demonstrated that pancreatic cancer cells express both TLR2 and TLR4 and that TLR2 stimulation enhances the frequency of CD133^+^ pancreatic CSCs, while TLR4 exerts the opposite effect, indicating that the balance of activation of the two receptors may regulate pancreatic cancer stemness [77]. Similarly, Wang and collaborators demonstrated that TLR2 expression is upregulated in glioma CSCs as compared to more differentiated glioma cells, and its activation increases CSC invasion and migration by enhancing the expression of MMP2 and 9 [78]. Finally, in ovarian cancer, TLR2 expressed on CD44^+^/MyD88^+^ CSCs promotes their self-renewal and induces the secretion of pro-inflammatory cytokines, thus actively contributing to tumor repair following injury induced by surgery or chemotherapy [79].

## 8. Effect of Chemo and Radiotherapy on TLR2 Activation

The activation of TLR2 signaling pathway in solid tumors may be induced by numerous DAMPs, such as different HSPs, HMGB1, fibrinogen, and hyaluronic acid fragments [80]. DAMP molecules are mainly represented by intracellular proteins that play a physiological role inside the cells. In particular conditions, DAMPs can be released, actively secreted, or exposed by some immune and non-immune cells, and they act as danger signals that stimulate an inflammatory response [81,82]. The TME contains DAMPs that are passively released by necrotic cells, which are often enriched in the inner core of the solid tumor mass as a consequence of hypoxia and acidic pH [83]. Moreover, inflammatory cells recruited in the tumor can actively secrete DAMP molecules through a non-canonical secretory pathway that involves the activation of caspase-1 and the inflammasome or a vesicle-mediated process. Finally, many cancer cells overexpress different DAMPs and can actively or passively secrete them in response to exogenous or endogenous stimuli such as hypoxia and several cytokines (such as TNF-α and IL-1), thus enhancing DAMP levels in the TME [83]. Although, physiologically, the role of DAMP-induced TLR signaling is to activate a pro-inflammatory immune response in order to induce tissue regeneration, their binding to TLR2 and other receptors on tumor cells may promote cancer progression [82]. One of the DAMPs that can stimulate TLR2 activation in the tumor milieu is HMGB1. In normal conditions, HMGB1 is a constitutively expressed protein primarily localized in the cell nucleus, where it binds DNA and transiently bends it, favoring nucleosome formation and contributing to DNA transcription, replication, and repair [84]. However, in neoplastic tissues, HMGB1 is actively secreted by inflammatory cells or tumor cells or passively released by dead cells, and it can activate its cognate receptors, including TLR2 [82]. We have previously demonstrated that breast CSCs actively secrete HMGB1, which fosters their own self-renewal through the activation of the TLR2 signaling pathway in an autocrine or paracrine manner [73] (Figure 3). Similarly, other groups demonstrated that cancer cells of different origin, such as those from malignant mesothelioma [85], breast [86], thyroid [87], and gastric [88] cancers, actively release HMGB1, which promotes their growth.

Importantly, recent data have demonstrated that DAMPs, and in particular HMGB1, can also be released by cancer cells undergoing immunogenic cell death (ICD) or necroptosis. ICD is a mechanism of induced cell death in which tumor cells expose calreticulin on their surface and release ATP or HMGB1 during apoptosis or necrosis. Instead, necroptosis is a programmed form of necrosis activated by different death receptors (such as TNF receptor 1, Fas, and TNF-related apoptosis-inducing ligand) and mediated by the receptor-interacting protein kinase (RIPK)3 and mixed lineage kinase domain-like protein (MLKL), which form the necrosome complex [89]. It is well known that both radiotherapy and chemotherapy may induce ICD or necroptosis in cancer cells [90,91], and therefore increase HMGB1 release in the TME [92]. This release of DAMPs may have detrimental effects on the efficacy of these anticancer therapies, since it can induce TLR2 activation on both neoplastic and immunosuppressive cells, thus promoting the generation of an immunosuppressive TME, tumorigenesis, and metastatic dissemination [22]. Indeed, HMGB1 is released in the TME after chemotherapy in breast and pancreatic cancer patients [93], and high HMGB1 levels have been associated with poor response to treatment in pancreatic and triple negative breast cancer patients [94,95]. Similarly, HMGB1 released by dying cells accelerates the development of pancreatic carcinoma metastasis following radiotherapy. This is a consequence of the dedifferentiation of CD133^−^ cancer cells into CD133^+^ CSCs mediated by TLR2 activation of the Hippo-Yes-associated protein (YAP) signaling pathway, suggesting that chemotherapy- and radiation-induced cancer cell death establishes a supporting niche for cancer cell stemness [96]. Moreover, the activation of the HMGB1/TLR2 signaling pathway induced by chemo- and radiotherapy promotes autophagy, which is a multistep mechanism for the recycling of misfolded proteins, lipids, damaged organelles, and other cellular components, which leads to their lysosomal degradation. Autophagy maintains metabolic and cellular homeostasis by providing energy production and the elimination of damaged materials, thus promoting cell survival [97]. Although in the early phases of neoplastic transformation autophagy may exert a tumor suppressive role by preserving cellular homeostasis, in overt tumors, it promotes disease progression by reducing ROS-induced metabolic stress and making nutrients available for cancer cell survival [89]. Moreover, autophagy can promote invasion, since it induces the release of TGF-β and, consequently, EMT in cancer cells, thus favoring metastasis [98]. Of note, the HMGB1/TLR2 signaling pathway may trigger a positive feedback loop that fosters cancer resistance to multiple treatments. Indeed, autophagy confers drug and irradiation resistance in several types of cancer and, in the meanwhile, it induces the release of additional HMGB1 in the TME, which can further promote autophagy [89]. However, it should be considered that HMGB1 may act as a checkpoint between autophagy and apoptosis, depending on the context. In fact, HMGB1 possesses three conserved cysteines whose modification regulates the activity of extracellular HMGB1. In particular, reduced HMGB1 activates autophagy, while its oxidized form induces apoptosis in neoplastic cells [97], suggesting that the HMGB1/TLR2 axis would be particularly active in promoting cancer growth in a reducing microenvironment [99]. Of note, many tumors express high levels of xCT (*SLC7A11*), which is the light chain of the redox state regulator cysteine/glutamate antiporter system Xc- [100,101], whose expression is particularly enriched on CSCs [76,102]. xCT activity regulates a redox cycle consisting of cystine import, its intracellular reduction to cysteine, which is then secreted and reoxidized to cystine in the extracellular environment. Therefore, xCT contributes to the maintenance of a reducing TME that may preserve HMGB1 ability to activate TLR2 [103]. We have previously demonstrated that xCT protects breast CSCs from oxidative stress and chemotherapy, and we have developed different anti-xCT vaccines that decrease tumor growth and metastasis in breast cancer mouse models [76,101,102,104,105]. We hypothesize that the ability of xCT immunotargeting to inhibit the HMGB1/TLR2 axis may contribute to its therapeutic activity and suggest that it would be worth developing combined therapies targeting these two mechanisms responsible for tumor survival and progression.

## 9. Interaction between TLR2-Expressing Cells and the Microbiota

In addition to DAMPs, TLR2 activation may be induced by PAMPs. Although this mechanism has been extensively studied in immune cells, recent evidence indicate that it may play a role in cancer cells as well [52,66]. Scientists are becoming increasingly aware of the existence and importance of a strict interplay between host cells and the microbiota that populates human tissues. Indeed, chronic infection by bacteria represents a risk factor for tumorigenesis, and the underlying mechanism is attributed to infection-induced inflammation. The major evidence of a link between pathogens and cancer come from tumors of the gastrointestinal system. Indeed, many studies identified different bacterial species that cause colon tumorigenesis both by activating TLRs on cancer cells and by tumor cell-extrinsic mechanisms that involve the stimulation of pro-tumorigenic effects in host immune cells [106,107]. The best known bacterial etiologic agent of gastric adenocarcinoma and colon cancer is represented by *Helicobacter pylori.* This pathogen promotes tumor onset and progression through multiple mechanisms, including TLR2 upregulation on intestinal epithelial cells and fibroblasts, with the subsequent TLR2-mediated stimulation of their proliferation and EMT through the activation of NF-κB and MAPKs, leading to the secretion of IL-6 [58,107]. Similarly, the presence of the anaerobic, Gram-negative *Fusobacterium nucleatum* (*F. nucleatum*) is abundant in colorectal cancer specimens and is linked to poor prognosis in patients. *F. nucleatum* is a normal oral commensal that can spread to different tissues under unhealthy conditions, acting as an opportunistic pathogen in different diseases [108]. In particular, *F. nucleatum* is enriched in Gal-GalNac expressing tumors, such as colorectal and breast cancers, where it promotes tumor progression by inducing TLR2/TLR4- and E-cadherin-dependent activation of NF-κB and Wnt pathways on cancer cells [108], by inhibiting NK and T cell activation through interaction with the T cell immunoglobulin and ITIM domain (TIGIT) inhibitory receptor [109], and by inducing TLR2-dependent Treg activation [110,111].

Apart from the pro-tumorigenic effect of single bacterial species, many studies have evidenced the presence of compositional shifts of the gut microbiota associated with colorectal cancer, with the appearance of pathogenic bacteria and the decrease of commensal species. The consequent dysbiosis may promote carcinogenesis through different mechanisms, including the alteration of cancer metabolism induced by bacteria metabolites, the alteration of the microenvironment, with the promotion of pro-inflammatory immune responses or the recruitment of immunosuppressive cells. Although several pathways are involved in this phenomenon, and the exact mechanisms have not been completely clarified, TLR2 may play a role by mediating both cancer cell-intrinsic and extrinsic mechanisms [106].

Other types of cancer, such as breast and lung cancer, have been associated with gut dysbiosis, since the gut microbiota can control cancer cell proliferation and metabolism through the secretion of metabolites that act as systemic hormones [112]. Moreover, although poorly characterized, the presence of a specific microbiota in other tissues such as the mammary gland and the lower respiratory tract has been recently demonstrated [113,114]. Even if the study of the breast and lung microbiota is still in its infancy, differences in their amount and composition between normal and neoplastic samples have been identified, suggesting that they may contribute to carcinogenesis [113,114]. Indeed, a link between pathogens and cancer progression is suggested by the fact that periodontitis, bacterial-induced mastitis in case of breast cancer, infections, and bacterial contamination after tumor surgery represent risk factors for metastasis development [66]. In particular, post-operative infectious complications occur in 15–40% of patients following cancer surgery, and they are associated with increased metastasis and decreased overall survival in different types of cancer [115]. In light of the central role exerted by TLR2 in cancer progression and in pathogen recognition, the increase of pathogens able to activate TLR2 on cancer cells and CSCs might contribute to the complex net of mechanisms underlying the pro-tumorigenic effect of particular bacteria.

Of note, it has been recently shown that chemotherapy induces dysbiosis in the gut, with a deep compositional and functional imbalance in the microbial community [116]. Similarly, neoadjuvant chemotherapy alters the composition of the mammary microbiota in patients affected by locally advanced breast cancer, increasing *Pseudomonas* and decreasing *Streptococcus* populations [117]. Moreover, most patients undergoing chemotherapy suffer from opportunistic infections, mostly from Gram-positive bacteria, and they display a diffusion of commensal bacteria to the tumor [118], which could further activate TLR2 on cancer cells. Therefore, TLR2 signaling may represent a good therapeutic target to improve the efficacy of anti-cancer therapies.

## 10. TLR2-Targeted Anti-Cancer Therapy

The activation of TLR2 on effector immune cells plays important roles in the immune response. Thus, TLR2 agonists have been proposed as adjuvants for anti-cancer vaccination [15]. In particular, TLR2 ligands have been used to activate DCs in vitro to improve antigen-specific T cell responses or as components of peptide-based vaccines. In this perspective, the covalent conjugation of the Pam_3_CSK_4_ TLR2 ligand to synthetic long peptides derived from human papillomavirus (HPV) type 16 resulted in a high presentation of the antigen by DCs, which are induced by the TLR2-independent uptake of the construct via clathrin- and caveolin-mediated mechanisms, and in the TLR2-dependent activation of DCs [119]. Thus, the conjugation of Pam_3_CSK_4_ to peptide vaccines improved their anti-cancer activity in preclinical models [120], and a vaccine composed of a synthetic TLR2 ligand (called Amplivant^®^) conjugated to HPV16 E6 long peptides is currently undergoing a phase I clinical trial in patients with HPV16 positive tumors or premalignant lesions (NCT02821494). Although other vaccines exploiting TLR2 agonists as adjuvants have been recently tested in different preclinical models of solid tumors, clinical data on their safety and effectiveness have not been provided so far [121,122], in contrast to agonists of other TLRs that are extensively tested in the clinic [123]. Actually, TLR2 is activated by two poly-TLR agonists currently used or tested in cancer treatment, i.e., a Food and Drug Administration-approved live attenuated *Mycobacterium bovis* preparation of bacillus of the Calmette–Guerin strain and CADI-05, a polyantigenic vaccine containing heat-killed *Mycobacterium indicus pranii* currently undergoing clinical investigation. However, since these two immunomodulators activate a wide range of TLRs, it is not possible to infer the contribution of TLR2 activation to their anticancer activity [124].

In light of the pro-tumorigenic and immunosuppressive functions of TLR2 discussed in the previous paragraphs, we suggest that TLR2 may act as a double-edge sword in cancer therapy and that the use of TLR2 agonists in cancer treatment should be carefully evaluated, since it could potentiate cancer cell growth and immunosuppressive mechanisms. A hint toward this hypothesis is conferred by the observation that a vaccine constituted by irradiated B16 melanoma cells engineered to express GM-CSF suppressed tumor growth in TLR2^−/−^ but not in wild-type mice [37]. Indeed, many evidences suggest that TLR2 inhibition may represent a valid strategy for combined anti-cancer therapies, as summarized in Figure 4. The feasibility of this approach is sustained by data from TLR2 knockout mice, showing that both inflammatory responses and the development of the acquired humoral response can occur in the absence of TLR2 [125]. Up to now, TLR2 inhibition in human patients has been tested in some clinical trials for the treatment of hematologic malignancies, using either monoclonal antibodies or small molecule inhibitors. A humanized IgG4 monoclonal antibody (OPN-305) targeting the TLR2 ligand-binding site, thereby preventing its heterodimerization with TLR1 or TLR6, was developed by Opsona Therapeutics. This antibody was well tolerated in healthy subjects [126], and it was tested in phase I/II clinical trials as a second or third-line monotherapy or combination treatment with azacytidine in patients affected by low or intermediate risk myelodysplastic syndrome, giving an overall response rate in terms of hematologic improvement of 50% (NCT02363491 and NCT03337451) [127]. Even better results in myelodysplastic syndrome were obtained using CX-01 (NCT02995655), which is a heparin derivative working as a TLR2/4 inhibitor (although not specific, as it also impairs the CXCL12/CXCR4 axis) [128]. Moreover, CX-01 was successfully used to treat patients with acute myeloid leukemia (AML) when combined with standard therapy (consisting of cytarabine and idarubicin), giving a complete remission rate of 92% in a pilot study (NCT02995655) and of 89% in a randomized phase II trial conducted in elderly patients (NCT02873338), suggesting that TLR2 antagonists may potentiate the efficacy of standard AML induction therapy [128].

Up to now, no clinical trial of TLR2 inhibition in patients suffering from solid cancers has been performed. However, data from preclinical models suggest that TLR2 inhibition may exert anti-cancer effects. Treatment of human head and neck squamous cell carcinoma (HNSCC) cells or patient-derived xenografts implanted in immunocompromised mice with the anti-TLR2 monoclonal antibody T2.5 significantly impaired their in vivo growth, suggesting that TLR2 targeting may improve the outcome of HNSCC patients. Since in the oral cavity TLR2 may be activated by components of the oral microbiota, and alterations of oral microbiota composition have been associated with oral squamous cell carcinoma [129], it is conceivable that the reduction or manipulation of the oral microbial load might have significant effects on the progression of oral HNSSC [130]. Similarly, the treatment of mice bearing human gastric cancer xenografts or of gp130^F/F^ genetically modified mice spontaneously developing gastric cancer with anti-TLR2 monoclonal antibodies (OPN-305 or its mouse counterpart OPN-301, respectively) significantly impaired neoplastic cell proliferation and induced their apoptosis, resulting in a 2.5 reduction of tumor growth as compared to control mice. These data indicate that TLR2 is a likely therapeutic target in advanced human gastric cancer and that TLR2-directed therapies might represent a new first- or second-line adjuvant treatment [57]. Of note, TLR2 blocking antibodies exert a double role by acting not only on cancer cells but also on host immune cells, reversing the tumor cell-induced immunosuppressive microenvironment and restoring the activity of anti-cancer cells such as cytotoxic T cells and M1 macrophages [131].

Apart from TLR2 neutralizing antibodies, new molecular inhibitors of TLR2 have been developed in the last years, thanks to the increasing interest on TLR2 inhibition as a potential therapeutic strategy for several non-cancerous diseases such as Parkinson’s disease and other synucleinopathies [132], renal graft dysfunction [133], and inflammatory disorders [134]. Although these inhibitors have not been tested in cancer therapy yet, they present interesting features. While some of them such as C29, *ortho*-vanillin, and AT5 are able to block both TLR2/1 and TLR2/6 heterodimers [134,135], others, such as CU-CPT22 and MMG-11, act selectively on the TLR2/1 heterodimer [136]. Therefore, if further studies lead to distinguishing the TLR2 heterodimers involved in cancer progression from those involved in protective immune responses, the use of these small molecule TLR2 inhibitors will allow inhibiting those TLR2 signaling pathways that exert a detrimental role on cancer progression, while sparing those that favor anti-tumor immune responses.

In any case, both TLR2-neutralizing antibodies and synthetic antagonists could represent new candidates for the development of combined therapies. An interesting approach could be represented by their combination with costimulatory agents that activate adaptive immunity, in order to enable long-term tumor rejection. This approach has been investigated in a mouse model of melanoma, where the combination of a TLR2-neutralizing antibody with cytosine-phosphate-guanine (CpG) oligodeoxynucleotides able to activate TLR9 and initiate anti-tumor immune responses acted synergistically to exert a potent anti-metastatic activity [137]. Most importantly, TLR2 inhibitors could be combined with chemotherapy or radiotherapy in order to prevent the pro-tumorigenic role exerted by DAMPs released following these treatments or by their induced dysbiosis. This hypothesis has been confirmed by some preclinical studies. For example, the combination of an anti-TLR2 monoclonal antibody with the cytotoxic agent gemcitabine synergistically inhibited the development of pulmonary metastases in a preclinical model of mouse melanoma [131]. Of note, TLR2 is implied in the development of mucositis, which is a common side effect of chemotherapy induced by the release of DAMPs by intestinal epithelial cells damaged by the chemotherapeutic drugs, leading to inflammation, mucosal damage, and barrier dysfunction [138]. Therefore, the administration of TLR2 inhibitors in combination with chemotherapy might also improve its side effects.

Another promising combination therapy consists in the concomitant inhibition of TLR2 and VEGF pathways, which has the potential to improve patient response and clinical outcomes. Indeed, TLR2 promotes VEGF-independent angiogenesis, thus contributing to the development of drug resistance to anti-VEGF therapy [139]. This possibility has been explored by using a monoclonal antibody targeting the endogenous TLR2 ligands 2-(ω-carboxyethyl)pyrrole (CEP) derivatives of ethanolamine phospholipids, which is a group of compounds produced by radical-induced oxidation of docosahexaenoate-containing lipids that has been implied in the stimulation of TLR2-dependent angiogenesis [140]. The administration of an anti-CEP monoclonal antibody enhanced the efficacy of the anti-VEGF monoclonal antibody bevacizumab in a mouse model of glioblastoma, suggesting that the inhibition of the two pathways may exert synergistic effects [139]. However, since CEP represents only one of the ligands able to activate TLR2 in the TME, we suggest that better results might be obtained by using a TLR2-inhibiting agent in combination with bevacizumab.

## 11. Conclusions

It is well known that cancer cells undergo an intense cross-talk with the immune system, the TME, and the microbiota, reprogramming them to escape the immune response and promote cancer progression. The role played by TLR2 in this complex interaction is ambivalent, since its ability to activate protective immune responses is counterbalanced by its immunosuppressive and pro-tumoral effects, which have prevented the diffusion of its agonists as anti-cancer drugs. We are firmly convinced that a deeper understanding of the multifaceted activity played by TLR2 in cancer cells, immune cells, and in the TME could lead to strong improvements in medical oncology, posing the foundations for the development of new combined anti-cancer treatments. Therefore, in this review, we have summarized and discussed what is currently known in the field, providing the reader with a comprehensive analysis of the pro- and anti-cancer activities played by TLR2. We think that this topic is gaining a renewed interest for oncologists and tumor immunologists, in light of the recent breakthrough in the field of tumor microbiota that made us more aware of its influence on carcinogenesis and response to treatments. Indeed, accumulating data are suggesting that TLR2 activation induced by many current anti-cancer therapies that mediate the release of DAMPs or alterations in PAMPs present in the microbiota is detrimental for patient outcome. Therefore, we suggest that these anti-cancer therapies would benefit from the association with drugs able to inhibit TLR2 signaling and prevent its pro-tumoral activities. Thus, TLR2 targeting would perfectly fits with the awareness that the future of oncology is finding the right combination of therapies that synergistically push and pull from different directions to maximize anti-cancer effectiveness. We are currently studying the combination of TLR2 targeting, chemotherapy, and immunotherapy in breast cancer preclinical models. The results from our studies, together with a deeper understanding of the effects exerted by microbiota dysregulation during chemotherapy, will clarify in the next few years whether TLR2 antagonists could find a central place in combination therapies with chemotherapy, immunotherapy, and anti-angiogenic treatments. This knowledge may lead to the development of clinical trials associating TLR2 antagonists with chemo and immunotherapies in solid cancers, holding promise for the design of new successful combination therapies.

## Figures and Tables

**Figure 1 ijms-21-09418-f001:**
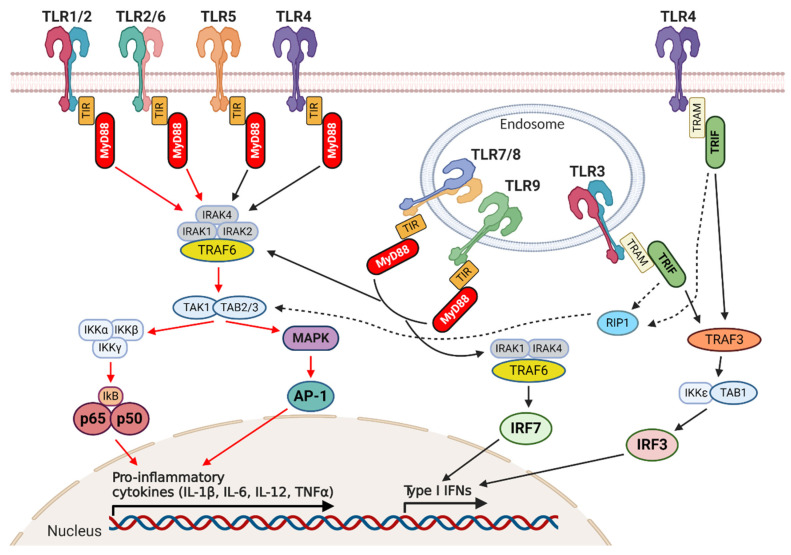
Toll-like receptor (TLR) signaling pathways. Schematic representation of TLR cellular location and their signaling pathways. TLR2, together with TLR1, TLR4, TLR5, and TLR6, is expressed on the external cellular membrane allowing the recognition of extracellular pathogen-associated molecular patterns (PAMPs)/damage-associated molecular patterns (DAMPs), while TLR3, TLR7, TLR8, and TLR9 are located in endosomal compartments where they bind nucleic acids. All the TLRs utilize the myeloid differentiation primary response protein 88 (MyD88) pathway except for TLR3, whose signaling depends on the TIR domain containing adaptor inducing IFN-β (TRIF) pathway. As indicated by the red arrows, TLR2 dimers activate the canonical MyD88-dependent pathway that, through the recruitment of the IL-1 receptor-associated kinase (IRAK) complex and TNF receptor associated factors (TRAF)6, leads to the activation of nuclear factor kappa-light-chain-enhancer of activated B cells (NF-κB), mitogen-activated protein kinase (MAPK), and activating protein-1 (AP-1), with the consequent induction of pro-inflammatory cytokine production. Intracellular TLRs are mainly involved in the type I interferon response. TLR7, TLR8, and TLR9 activate interferon regulatory factor 7 (IRF7) through the recruitment of IRAK and TRAF6. Since their signaling is initiated by MyD88, they can also activate the canonical pathway described above. Instead, the TRIF-dependent pathway, used by TLR3, or in some cases also by TLR4, leads to the activation of TRAF3 and consequently of IRF3, resulting in the induction of type I interferon (IFN) production. All arrows indicate activation: red arrows indicate the TLR2 signaling pathway, black and dashed arrows indicate all the other signaling pathways. Created with BioRender.com.

**Figure 2 ijms-21-09418-f002:**
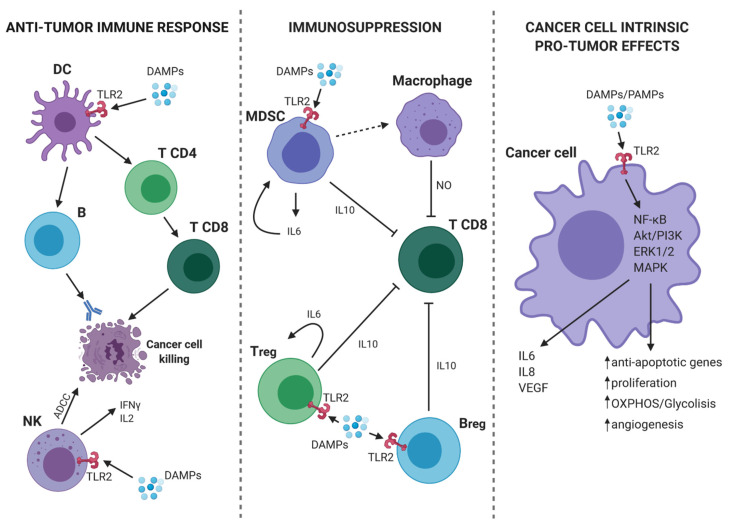
TLR2-mediated pro- and anti-tumor mechanisms. (On the left) In tumor microenvironment (TME), the presence of DAMPs activates dendritic cells (DCs) through TLR2. Consequently, DCs present the antigen and initiate a specific anti-tumor T cell response. At the same time, TLR2 expressed on Natural Killer (NK) cells mediates their activation and the induction of antibody-dependent cell-mediated cytotoxicity (ADCC) against cancer cells as well as the release of IFN-γ and interleukin (IL)-2. (On the center) TLR2 activation on myeloid-derived suppressor cells (MDSCs), Bregs and Tregs leads to an immunosuppressive effect. Indeed, MDSCs can induce macrophages to release nitric oxide (NO) that suppresses the activity of T CD8^+^ lymphocytes. Moreover, activated MDSCs, Bregs and T regulatory cells (Tregs), release IL-6 and IL-10 with the consequent synergic inhibitory effect on T CD8^+^ cells. (On the right) TLR2 expressed on cancer cells can bind DAMPs/PAMPs and activate intracellular signaling, such as nuclear factor kappa-light-chain-enhancer of activated B cells (NF-κB), Akt/PI3K, extracellular signal-regulated kinase (ERK)1/2, and MAPK. The consequence is a pro-tumor effect due to the release of IL-6, IL-8, VEGF, and the promotion of proliferation, cell metabolism, angiogenesis, and protection from apoptosis. Continous and dashed arrows indicate activation or secretion. T-arrows indicate cellular inhibition. Created with BioRender.com.

**Figure 3 ijms-21-09418-f003:**
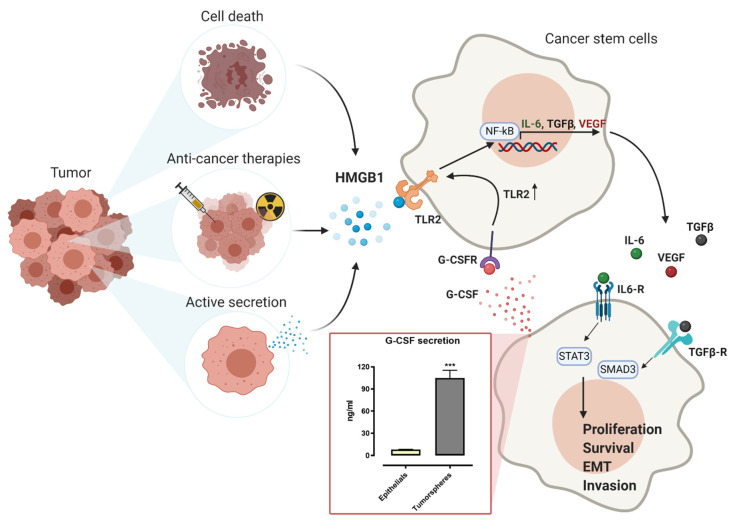
The high-mobility group box protein 1 (HMGB1)/TLR2 axis promotes cancer stem cell (CSC) self-renewal, proliferation, and invasion. TLR2 is expressed on many tumors, in particular on CSCs that release granulocyte colony-stimulating factor (G-CSF), as demonstrated by ELISA analysis of G-CSF production in TUBO epithelial breast cancer cells or their CSC-enriched tumorspheres (the graph shows results from three independent experiments. ***, *p* < 0.001, Student’s t test). In turn, G-CSF is an inducer of TLR2 expression. Cancer cell death and aggressive therapies, such as radio- and chemotherapies, lead to the release of HMGB1 in the extracellular space. In addition, HMGB1 can be also actively secreted by cancer cells. TLR2 binds HMBG1 and activates the transcription of pro-tumoral factors, such as IL-6, TGF-β, and VEGF, which act in an autocrine and paracrine manner, leading to cancer cell survival, proliferation, epithelial to mesenchymal transition (EMT), and invasion. All arrows indicate activation or secretion. Created with BioRender.com.

**Figure 4 ijms-21-09418-f004:**
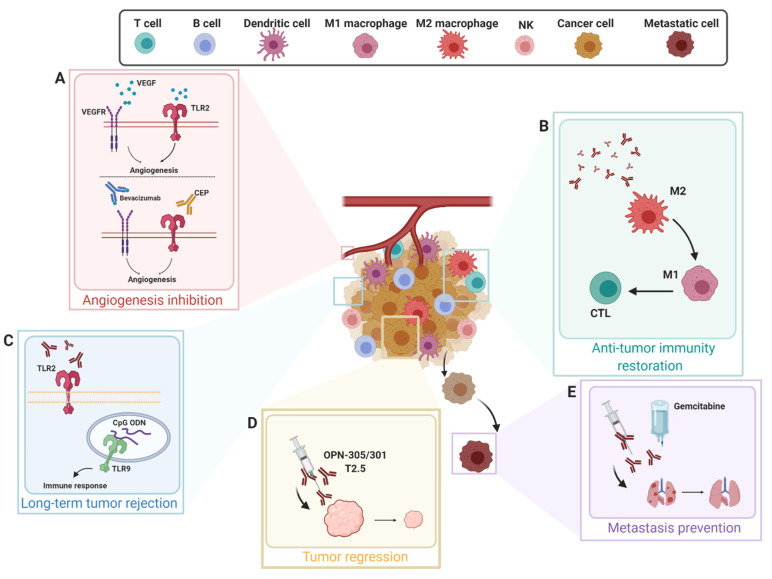
TLR-2 Targeted Anti-Tumor Therapies. TLR2 inhibition has an impact in hampering many aspects of tumor progression. (**A**) The engagement of TLR2 ligands such as 2-(ω-carboxyethyl)pyrrole (CEP) promotes both VEGF-therapy resistance and angiogenesis. The combination of anti-TLR2 and the anti-VEGF (Bevacizumab) monoclonal antibodies act synergistically to enhance the blocking of VEGF-independent angiogenesis. (**B**) Anti-TLR2 monoclonal antibodies act on both cancer cells and immune cells within the TME. As such, the anti-tumor ability of immune cells, such as cytotoxic T lymphocytes (CTLs) and M1 macrophages, is restored achieving a strong anti-tumor effect. (**C**) TLR2-neutralizing antibodies in combination with co-stimulatory agents activate anti-tumor immunity. For instance, the combination of cytosine-phosphate-guanine (CpG) oligodeoxynucleotides (ODN), which activate TLR9 and initiate an anti-tumor immune response, and anti-TLR2 antibodies elicit a strong immune response, finally triggering both a long-term tumor rejection and metastasis prevention. (**D**,**E**) Anti-TLR2 monoclonal antibodies are able to brake tumor progression and metastatization, enhancing the effect of standard chemotherapy in preclinical mouse cancer models. All arrows, black and curved, indicate different steps of a process. Created with BioRender.com.

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
