# Peer review of "Toll-Like Receptor 2 at the Crossroad between Cancer Cells, the Immune System, and the Microbiota"

_ijms, 2020, doi:10.3390/ijms21249418_

Round 1

Reviewer 1 Report

Dear Editor, we read with interest the review by Di Lorenzo and colleagues. 

Recent years have seen a growing interest in Toll-like receptors (TLRs) in oncology since TLRs represent are major components of the innate immune system that recognize the conserved molecular structures of pathogens (pathogen-associated molecular patterns; PAMPs). Of note, TLR binding triggers the expression of several downstream kinases, leading to the induction of key pro-inflammatory mediators. This results in the activation of both the innate immune response, as well as the adaptive immune response (antigen presentation, maturation of the dendritic cells, etc.). In consequence of their ability to enhance the specific and nonspecific immune reactions of an organism, TLR agonists have been explored in the therapy of infectious diseases and, as adjuvants, in the therapy of malignancies. However, to date, TLRs have had the opposite effects on tumor progression. On the one hand, TLR ligands can suppress tumor growth. On the other hand, TLR agonists can promote the survival of malignant cells and increase their resistance to chemotherapy. 

Since this topic is really important in oncology research, the paper addresses a timely subject. 

The manuscript is well written and organized. 

Figures and tables are comprehensive and clear.

The introduction explains in a clear and coherent manner the background of this study.

We suggest the following modifications:

  • Introduction section: although the authors correctly included important papers in this setting, we believe a couple of studies should be cited within the introduction (10.3389/fphar.2018.0087810.3390/cancers12082308), only for a matter of consistency. We think it might be useful to introduce the topic of this interesting study. 
  • Other sections: interesting and timely parts. Of note, the authors should expand the 5. and 6. sections, including a more personal perspective to reflect on. For example, they could answer the following questions – in order to facilitate the understanding of this complex topic to readers: what potential does this study hold? What are the knowledge gaps and how do researchers tackle them? How do you see this area unfolding in the next 5 years? We think it would be extremely interesting for the readers, especially considering the importance of this topic in medical oncology. 

However, we think the authors should be acknowledged for their work. 

We believe this article is suitable for publication in the journal although minor revisions are needed. The main strengths of this paper are that it addresses an interesting and very timely question and provides a clear answer, with some limitations. Certainly, the study is limited by its nature of narrative review.

We suggest some changes and the addition of some references for a matter of consistency. 

Author Response

Recent years have seen a growing interest in Toll-like receptors (TLRs) in oncology since TLRs represent are major components of the innate immune system that recognize the conserved molecular structures of pathogens (pathogen-associated molecular patterns; PAMPs). Of note, TLR binding triggers the expression of several downstream kinases, leading to the induction of key pro-inflammatory mediators. This results in the activation of both the innate immune response, as well as the adaptive immune response (antigen presentation, maturation of the dendritic cells, etc.). In consequence of their ability to enhance the specific and nonspecific immune reactions of an organism, TLR agonists have been explored in the therapy of infectious diseases and, as adjuvants, in the therapy of malignancies. However, to date, TLRs have had the opposite effects on tumor progression. On the one hand, TLR ligands can suppress tumor growth. On the other hand, TLR agonists can promote the survival of malignant cells and increase their resistance to chemotherapy. 

Since this topic is really important in oncology research, the paper addresses a timely subject. 

The manuscript is well written and organized. 

Figures and tables are comprehensive and clear.

The introduction explains in a clear and coherent manner the background of this study.

We thank the Reviewer for the comments.

We suggest the following modifications:

  • Introduction section: although the authors correctly included important papers in this setting, we believe a couple of studies should be cited within the introduction (10.3389/fphar.2018.00878 ; 10.3390/cancers12082308), only for a matter of consistency. We think it might be useful to introduce the topic of this interesting study. 

RESPONSE: As suggested by the Reviewer, we have added the reference 10.3389/fphar.2018.00878 in both the introduction and in section 2. We have not added the reference 10.3390/cancers12082308 because it refers to the evolution of the models of cholangiocarcinoma, a type of tumor that we have not analyzed in the paper, and does not mention TLRs.

  • Other sections: interesting and timely parts. Of note, the authors should expand the 5. and 6. sections, including a more personal perspective to reflect on. For example, they could answer the following questions – in order to facilitate the understanding of this complex topic to readers: what potential does this study hold? What are the knowledge gaps and how do researchers tackle them? How do you see this area unfolding in the next 5 years? We think it would be extremely interesting for the readers, especially considering the importance of this topic in medical oncology. 

RESPONSE: As suggested by the Reviewer, we have now extended sections 5 and 6, discussing the strengths and drawbacks of the papers that in our opinion provided the most important advancements in the field. Moreover, we have added in the Conclusion section our personal perspectives on the topic and on its evolution in the next few years, providing an overview on how we are currently experimentally addressing the open questions.

However, we think the authors should be acknowledged for their work. 

We believe this article is suitable for publication in the journal although minor revisions are needed. The main strengths of this paper are that it addresses an interesting and very timely question and provides a clear answer, with some limitations. Certainly, the study is limited by its nature of narrative review.

We suggest some changes and the addition of some references for a matter of consistency. 

We thank the Reviewer for the positive comments.

Reviewer 2 Report

Comments to Author

   The authors describe function of Toll-like receptor (TLR)2 in normal and pathological conditions. The content of the manuscript is solid and gives an insight into mechanisms of cancer development and treatment. A thorough search has been sufficiently summarized and discussed with proper explanation.

I would recommend the authors to change the following points.

The sub-titles of individual figure are not informative. Furthermore, figures are not explained in the figure legend at all. Please add the explanations for the figures in the figure legends.

Page 5 line 199, page 6, line 225, and others. NF-kB should be NF-κB.

Page 7, line 274, “sterol regulatory element-binding protein-2”  The font-size seems to be big.

Author Response

The authors describe function of Toll-like receptor (TLR)2 in normal and pathological conditions. The content of the manuscript is solid and gives an insight into mechanisms of cancer development and treatment. A thorough search has been sufficiently summarized and discussed with proper explanation.

We thank the Reviewer for the positive comments.

I would recommend the authors to change the following points.

The sub-titles of individual figure are not informative. Furthermore, figures are not explained in the figure legend at all. Please add the explanations for the figures in the figure legends.

RESPONSE: As suggested by the Reviewer, we have modified the sub-titles of individual figures in order to make them more informative, and we have extended the explanation of the figures in the figure legends, which we have now moved from the “Figure legend” section to each figure.

Page 5 line 199, page 6, line 225, and others. NF-kB should be NF-κB.

RESPONSE: Thanks for this observation, we have modified NF-κB accordingly throughout the manuscripts.  

Page 7, line 274, “sterol regulatory element-binding protein-2”  The font-size seems to be big.

RESPONSE: As suggested, we have modified the font.